# Aptamer-Based Technologies for Parasite Detection

**DOI:** 10.3390/s23020562

**Published:** 2023-01-04

**Authors:** Noah Emerson Brosseau, Isabelle Vallée, Anne Mayer-Scholl, Momar Ndao, Grégory Karadjian

**Affiliations:** 1UMR BIPAR, Anses, Laboratoire de Santé Animale, INRAE, Ecole Nationale Vétérinaire d’Alfort, 94700 Maisons-Alfort, France; 2Infectious Diseases and Immunity in Global Health (IDIGH) Program, Research Institute of McGill University Health Centre, Montreal, QC H4A 3J1, Canada; 3Department of Biological Safety, German Federal Institute for Risk Assessment, 10589 Berlin, Germany

**Keywords:** biosensor, aptamer, parasite, diagnostics, aptasensor, detection

## Abstract

Centuries of scientific breakthroughs have brought us closer to understanding and managing the spread of parasitic diseases. Despite ongoing technological advancements in the detection, treatment, and control of parasitic illnesses, their effects on animal and human health remain a major concern worldwide. Aptamers are single-stranded oligonucleotides whose unique three-dimensional structures enable them to interact with high specificity and affinity to a wide range of targets. In recent decades, aptamers have emerged as attractive alternatives to antibodies as therapeutic and diagnostic agents. Due to their superior stability, reusability, and modifiability, aptamers have proven to be effective bioreceptors for the detection of toxins, contaminants, biomarkers, whole cells, pathogens, and others. As such, they have been integrated into a variety of electrochemical, fluorescence, and optical biosensors to effectively detect whole parasites and their proteins. This review offers a summary of the various types of parasite-specific aptamer-based biosensors, their general mechanisms and their performance.

## 1. Introduction

Parasitic illnesses continue to contribute significantly to the global burden that infectious diseases have on humanity. While developing countries are most heavily afflicted by cases of malaria, cysticercosis, and schistosomiasis, cases of parasitic disease can also be found in the world’s wealthiest nations, where food and waterborne illnesses such as cryptosporidiosis, giardiasis, toxoplasmosis, echinococcosis and trichinellosis are continual causes of concern for public health authorities [1]. In 2010, the World Health Organization (WHO) reported an estimated 137,000 deaths, and 15 million disability-adjusted life years (DALYs) caused by foodborne parasitic diseases [2,3,4,5]. Furthermore, vector-borne diseases caused by parasites, bacteria, and viruses result in an estimated 700,000 deaths every year, with 400,000 being caused by malaria alone [6]. In addition to their visibly debilitating effects on human and animal health, the results of parasite illness can also be seen at the economic level, where worker and livestock productivity is reduced as a consequence of crippling morbidity and death [7]. Despite the many scientific breakthroughs in the prevention, diagnosis, treatment, and control of infectious diseases, the remarkable ability of infectious pathogens to evolve and adapt to changing environments and pressures has made their eradication a seemingly impossible endeavor. Luckily, in the face of perpetual adversity, man has showcased his equally remarkable ability to observe, learn, and innovate.

Along with their critical role in providing proper and timely treatment to patients and animals, accurate and robust diagnostics are equally essential in acquiring crucial surveillance data for developing effective public health strategies [8]. While recent accomplishments in the surveillance and control of parasitic diseases can be partly attributed to advances in rapid detection and treatment, conventional parasitological, serological, and molecular detection methods are still considered standard practice. Unfortunately, these methods can be highly cumbersome, expensive, and difficult to perform in resource-limited settings. Furthermore, some of these modes of detection struggle with distinguishing between past, latent, acute, and reactivated infections, making it difficult to implement proper treatment regimens [9,10]. As a result, the continued development and improvement of point-of-care (POC) diagnostic tools such as rapid diagnostic tests (RDTs) have demonstrated great potential in resource-limited settings and hold much promise in disease diagnostics, active surveillance, and treatment [11,12]. While a wide array of rapid immunochromatographic tests have been developed for protozoan and helminth infections, potential cross-reactivity in cases of co-infection can lead to reduced test specificity. In light of this, the current state of POC testing for parasitic diseases mostly allows for preliminary diagnostics, whereby initial findings are used to determine if further parasitological diagnostics are required [13].

## 2. Biosensors

Biosensors are compact and portable analytical devices capable of rapid diagnostic testing that convert biological and/or chemical reactions into signals proportional to an analyte’s concentration (Figure 1). In addition to monitoring disease treatment and progression, biosensors have also found uses in many areas of drug discovery, food safety, and environmental monitoring [14,15]. Classically, biosensors employ bioreceptors or molecular recognition elements (MREs) that function to provide analyte specificity to the biosensor. Several classes of MREs exist, each with their own advantages and limitations in terms of biosensing selectivity, sensitivity, reproducibility, and reusability [16]. Such recognition elements may be derived from naturally occurring materials (e.g., enzymes, antibodies, nucleic acids, cell receptors, microorganisms), that take advantage of naturally-evolved interactions, or synthetic biorecognition elements (e.g., synthetic peptides and receptors, imprinted polymers), that can be engineered to mimic the physiological activities of natural elements [17]. Despite the wide variety of options, antibodies remain a staple in biosensing platforms, specifically for the detection of pathogens [16,18].

In order to achieve analyte detection, the MRE of choice is first immobilized onto the surface of a transducer before being challenged with the target analyte. Following bio-recognition, an event described as the interaction between bioreceptor and analyte, a transducer converts the reaction into a measurable signal [14]. The transducer is a vital element of any successful biosensor and may be electrochemical, optical, mass-based (piezoelectric), magnetic, thermometric or micromechanical [17,19]. As with the category of MRE, the transduction system heavily influences the biosensors performance and is chosen based on the type of signal that is required. Finally, ongoing advances in nanobiotechnology offer new approaches into biosensor development. Nanomaterials such as gold nanoparticles, carbon nanotubes (CNTs), magnetic nanoparticles, quantum dots, and graphene are some examples of nanostructures being integrated into biosensor technologies to enhance sensitivity and specificity [20].

## 3. Aptamers and SELEX

In 1990, two independent research groups stumbled upon the remarkable discovery of what Andrew Ellington and Jack Szostak termed ‘aptamers’, from the Latin ‘aptus’, to fit [21,22]. These short ssDNA or RNA oligonucleotides, varying in length from 20–100 nucleotides, displayed an impressive ability to bind various targets with high affinity and specificity. Their highly unique tertiary structures, dictated by their sequence diversity, grant aptamers with particular three-dimensional motifs such as loops, bulges, hairpins, G-quadruplexes, and pseudoknots [23]. By means of non-covalent interactions such as van der Waals, electrostatic forces, and hydrogen bonding, aptamers can be used as molecular recognition elements (MREs) to capture many types of small molecules (metal ions, hormones, antibiotics), macro-molecules (antibodies, antigens, viruses), live cells, and even whole microorganisms [24,25,26].

During the same time, while studying the interaction between the bacteriophage T4 DNA polymerase (gp43) and its encoding mRNA, Craig Tuerk and Larry Gold developed a novel method for the selection of preferred binding sequences from a highly variable population. This procedure, which they termed Systematic Evolution of Ligands by Exponential Enrichment (SELEX), was founded on the general mechanisms of evolution, whereby forces of variation, selection, and replication are implemented to gradually select molecular species specific to a target of interest (Figure 2) [22]. In conventional SELEX, a nucleic acid library consisting of approximately 10^15^ unique ssDNA or RNA sequences is incubated with a target molecule. After removing non-binding sequences, target-specific species are retrieved and PCR-amplified (RT-PCR amplified in the case of RNA aptamers). Following amplification, a crucial step of ssDNA generation from dsDNA PCR products can be achieved by means of asymmetric PCR, magnetic bead separation by biotin-streptavidin interaction, strand separation under denaturing conditions, and lambda exonuclease digestion [27]. This pool of sequences is subsequently subjected to additional rounds of selection with gradually increasing degrees of stringency to ensure the successful selection of high-affinity binders.

### Aptamers as Diagnostic Reagents

Since the pioneering publications of aptamer-based biosensors [28,29], a variety of diagnostic ‘aptasensors’ have been designed. These biosensors, which employ aptamers as MREs, circumvent the costly and complicated production of antibody bioreceptors. Furthermore, in contrast with their protein counterparts, aptamers exhibit high thermal stability, are easily modifiable, and can be easily conjugated to a wide range of reporter molecules [30,31,32]. Despite their obvious advantages, a large portion of biosensors still rely on antibodies for analyte detection. Prone to heat-induced damage, this is especially troublesome in warm, tropical climates, where parasitic diseases flourish. Since the introduction of a thrombin-detecting electrochemical-based aptasensor [33,34], their application in the field of infectious disease diagnostics has expanded towards tuberculosis (TB) [35], *Listeria monocytogenes* [36], dengue virus [37], and HIV [38] detection, among many others [39].

In this paper, we focus on the various applications of aptasensors and aptamer-based detection strategies in the field of parasitology. From detection in humans and animals to detection in environmental samples and insect vectors, the aptasensors described below showcase only a few potential uses. Despite the wide variety of parasites targeted by conventional antibody-based tests, the only parasites targeted by aptamer-based technologies thus far have been *Plasmodium* spp., *Trypanosoma* spp., *Leishmania* spp., *Cryptosporidium parvum*, *Toxoplasma gondii*, and *Trichomonas vaginalis* (Table 1). Used in conjunction with nanomaterials, these aptamer-based sensors employ electrochemical, fluorescent, colorimetric, and optical modes of detection.

## 4. Aptasensors for Parasite Detection

### 4.1. Electrochemical Aptasensors

Due to their portability, high sensitivity and specificity, quick response, and relative low cost, electrochemical aptasensors have gained much interest since their introduction in 2004 [33,34]. Since then, various aptasensors operating by voltammetric, amperometric, impedimetric, and potentiometric modes of transduction have been described [74]. Typical electrochemical aptasensors use an electrode surface, preferably gold (Au) or carbon-based, as the substrate onto which a redox probe-labeled aptamer, specific for a target of interest, is immobilized (Figure 3). In theory, aptamer-target binding induces a conformational change, transporting the probe closer to the electrode surface in the commonly used “signal-on” architecture [75,76]. This was demonstrated in an earlier work, where aqueous potassium ions (K^+^) converted an aptamer sequence from a loose coil to a compact G-quadruplex conformation resulting in electrochemical signaling [76]. Alternatively, a “signal-off” aptasensor, whose conformational change carries the probe further from the electrode, was used to measure the loss of redox current following thrombin recognition in blood serum samples [77]. Finally, an aptasensor featuring both “signal-on” and “signal-off” elements of signal amplification was developed for adenosine triphosphate (ATP) recognition [78]. 

#### 4.1.1. *Plasmodium* spp.

The first electrochemical aptasensor for *Plasmodium* detection was developed in 2012, targeting the well-known *Plasmodium* lactate dehydrogenase (pLDH) biomarker [56]. In comparison to its mammalian counterparts, protozoan LDHs display major structural differences, a feature that has been exploited to develop highly selective drugs and diagnostic systems [80,81]. Following its selection, aptamer pL1 was 5′-thiol-modified and integrated into a gold electrode-based aptasensor. pL1-pLDH binding was monitored by electrochemical impedance spectroscopy (EIS) and charge transfer resistance (R_CT_) was measured at the electrode surface to yield detection limits of 108.5 fM and 120.1 fM for *Plasmodium vivax* LDH (*Pv*LDH) and *Plasmodium falciparum* LDH (*Pf*LDH), respectively. When challenged with blood samples of infected patients, the aptasensor could detect pLDH corresponding to 100 parasites/µL [56]. Furthermore, enzyme-linked oligonucleotide assay (ELONA) and electrophoretic mobility shift assay (EMSA) displayed pL1′s superior affinity to *Pv*LDH and *Pf*LDH compared to another *Pf*LDH-specific aptamer, 2008s [46].

The additional discovery of a *Pf*LDH-specific ssDNA aptamer, termed P38, brought about the production of a graphene-oxide (GO) plated aptamer-electrode capable of probing *Pf*LDH [51,52]. With the addition of a NAD/NADH redox probe, lactate oxidation and *Pf*LDH activity at the aptamer-electrode surface was monitored electrochemically. By measuring reduced cofactor NADH in human serum samples, *Pf*LDH was detected at concentrations as low as 0.5 fM. Additionally, the aptasensor retained 62% of its functionality after storage at ambient temperature with a relative humidity of 70% to 90% [51]. 

Using the 2008s aptamer previously described by Cheung et al. as a recognition molecule, the group of Figueroa-Miranda et al. developed a label-free electrochemical impedance aptasensor for *Pf*LDH detection. Once immobilized, aptamer-*Pf*LDH based modifications at the electrode surface were evaluated by EIS using a ferri/ferro-cyanide redox probe. With this strategy, *Pf*LDH in 10-fold diluted human serum was detected at concentrations as low as 0.84 pM. In order to test the aptasensor’s stability, urea was used as a denaturation agent to disrupt the aptamer-protein interaction. Following three consecutive rounds of treatment, the sensor could still detect 1 nM of *Pf*LDH [40].

Aptamer 2008s was further tested using two independent transduction principles. EIS and surface plasmon polariton (SPP) were employed with a 2008s-modified gold electrode to monitor the presence of *Pf*LDH. Complementary signals generated by EIS and SPP-based transduction allowed for the detection of *Pf*LDH at concentrations of 1 pM to 100 nM and 10 nM to 1 µM, respectively [41].

Similarly, an ssDNA aptamer entitled NG3 was selected against *Plasmodium falciparum* glutamate dehydrogenase (*Pf*GDH), a mitochondrial enzyme and potential malaria biomarker due to its unique structure, sequence, kinetics, and absence in healthy host red blood cells [54,82,83]. With this aptamer, a capacitive aptasensor operating by a non-Faradaic mode of EIS transduction was developed. Once thiolated and fixed to the gold electrode, NG3-*Pf*GDH binding, along with the subsequent displacement of water molecules and ions from the electrode surface, resulted in changes in electron transfer equivalent to 0.77 pM of *Pf*GDH in serum samples. Additionally, interference from *Plasmodium falciparum* lactate dehydrogenase (*Pf*LDH) and histidine rich protein-II (*Pf*HRP-II), common malaria biomarkers, was negligible [54].

An additional study exploited the species specificity of *Pf*HRP-II to generate an ssDNA aptamer (B4) with an equilibrium dissociation constant (*K*_D_) of 1.32 µM. Using an impedimetric method to study *Pf*HRP-II adsorption, a B4-decorated gold electrode was fabricated. By measuring the charge transfer resistance (R_CT_) during B4-mediated *Pf*HRP-II recognition events, the resulting electrochemical aptasensor was capable of detecting at least 3.15 pM of *Pf*HRP-II [55]. 

*Pf*HRP-II was further investigated by the group of Tanner et al. to select an ssDNA aptamer after 21 rounds of SELEX. Compared to other candidates, aptamer 2106s displayed high affinity and specificity to *Pf*HRP-II. SPR estimated a *K*_D_ of 29.53 nM towards *Pf*HRP-II and no significant binding interaction with *Pf*LDH. An electrochemical aptamer-based biosensor was constructed by fixing methylene blue (MB)-conjugated 2106s to a gold surface electrode. The change in Faradaic current between the MB redox tag and the gold surface, governed by the aptamer’s change in conformation upon target binding, was monitored to determine the *K*_D_ of the 2106s aptasensor towards *Pf*HRP-II. In PBS, a *K*_D_ and limit of detection (LOD) of 51.84 nM and 2.47 nM were estimated, respectively. In diluted human serum spiked with *Pf*HRP-II, a *K*_D_ and LOD of 22.59 nM and 3.73 nM were determined, respectively [84].

Both pLDH and HRP-II have proven to be important biomarkers of malaria infection. While pLDH is expressed in all species of malaria, HRP-II is expressed exclusively by *P. falciparum* parasites [82,85]. In light of this, a flexible multielectrode array (flex-MEA), integrating the previously described aptamers 2008s, pL1, LDHp11, and 2106s, was fabricated for multi-target biosensing to discriminate between *P. falciparum* and *P. vivax* infections [86]. In blood samples spiked with biomarker proteins, changes in electric currents, induced by aptamer-protein binding, resulted in the detection of *Pf*LDH, *Pv*LDH, and HRP-II, by LDHp11, 2008s, and 2106s with LODs of 1.80 fM, 0.42 pM, and 0.15 pM, respectively. In parasite-infected whole blood samples, the LODs were estimated at 0.001% parasitemia (50 parasites/µL) with 93.3%, 100%, and 100% sensitivities, respectively, for 2008s, pL1, and 2106s [86].

#### 4.1.2. *Leishmania infantum*

Despite their potential as effective point-of-care diagnostic tools, electrochemical aptasensors targeting *Leishmania* spp. have not yet been explored in great detail. In contrast, several visceral leishmaniasis (VL) antibody-detecting RDTs targeting rK39 and rKE16 antigens are commercially available. However, while these tests are generally highly specific and sensitive, variation in product performances, attributed to parasite diversity, different antibody concentrations, and/or the biological fluid being assayed, have been described [87,88]. Additionally, the RDT DiaMed-IT-Leish displayed reduced sensitivity in cases of HIV/VL co-infection, [89]. In their work, Moreno et al. have presented a biosensor to detect *Leishmania infantum* kinetoplastid membrane protein 11 (LiKMP-11), a potential regulator of parasite mobility and host cell attachment [61]. After 10 rounds of SELEX, a pool of LiKMP-11 specific sequences, termed SELK10, was generated using a novel colloidal gold-based approach [90]. Following LiKMP-11 conjugation, gold nanoparticles were electro-deposited onto a screen-printed gold microelectrode, incubated with digoxigenin-labeled SELK10, and incubated with horseradish peroxidase (HRP)-labeled anti-digoxigenin antibody. Evaluating the HRP-mediated reduction in hydrogen peroxide, LiKMP-11 could be detected at concentrations as low as 2.27 µM [61].

#### 4.1.3. *Trypanosoma brucei*

As with leishmaniasis, research into aptamer-based diagnostic tools for trypanosomiasis remains rather underdeveloped. With that being said, several aptamers targeting the variable surface glycoproteins (VSGs) of *T. brucei* have been isolated. Coating the trypanosome surface, VSGs are targets of frequent antigenic variation, a mechanism that effectively undermines the development of a host antibody response, making vaccine development seemingly impossible [91,92,93]. RNA 2-16, an aptamer specific to live infective bloodstream stage African trypanosomes, was monitored with in situ fluorescence microscopy and localised to the parasite’s flagellar pocket, an invagination at the base of the flagellum with functions in exo- and endocytosis [94,95]. In order to investigate the potential ‘piggy-back’ carrier function of aptamers for therapeutic means, fluorophore-coupled anti-biotin antibodies were incubated with live trypanosomes in the presence of biotin-modified RNA aptamers. Remarkably, the endosomal transport of anti-biotin antibody from the flagellar pocket to the lysosome was confirmed and the mechanism further investigated in colocalization experiments with transferrin, whose results suggest receptor-mediated internalization [96]. In order to confer serum stability to the aptamer, 2′F-substituted pyrimidine nucleotides (2′F’dUTP/dCTP) were co-transcriptionally incorporated. While this led to reduced binding affinity, RNA 2-16 had benefited from an improved serum stability of up to 3.4 days with no changes in its characteristic binding to the flagellar pocket and eventual transport to the lysosome [97]. 

An additional study produced 2′F-modified RNA aptamers with high stability in mouse serum and affinities for several VSG variants in the subnanomolar range, suggesting the possible recognition of a structurally conserved domain. In fluorescence labeling experiments, biotin-conjugated aptamers displayed an ability to direct fluorophore-coupled anti-biotin antibodies to the surface of live trypanosomes [98]. A specific aptamer, entitled cl57, was further integrated into a single-walled carbon nanotube (SWCNT) based aptasensor for the potentiometric detection of VSGs in blood samples. When exposed to gradually increasing concentrations of VSG, the nanotube-tethered aptamer captures the target and undergoes a conformational change resulting in a switch of the SWCNT surface charge. The subsequent changes in electromotive force (EMF) readings indicated the detection of VSG concentrations as low as 10 pM. Finally, this aptasensor’s functionality could be recovered following a treatment with 2 M NaCl and washing with deionized water [65].

#### 4.1.4. *Cryptosporidium parvum*

A food and water-borne parasite, *Cryptosporidium* spp. infection is best prevented by monitoring its presence in fresh produce and frequently accessed bodies of water. However, the analysis of environmental oocysts is tedious, consisting of centrifugation, filtration, purification, and staining [99,100,101]. To detect *Cryptosporidium parvum* in pineapple and mango concentrates, an aptamer-based biosensor was fabricated using a gold nanoparticle-modified screen-printed carbon electrode (GNP-SPCE) with thiolated ssDNA capture probes. Of the aptamers tested, sequence R4-6 exhibited the highest binding affinity for the whole oocyst target with an estimated limit of detection (LOD) of approximately 100 oocysts [67]. The median infectious dose of *C. parvum* ranges from less than 30 to over 1000 oocysts [102]. An increase in this aptasensors sensitivity could therefore result in a promising alternative to conventional detection methods. Therefore, aptamer R4-6 was integrated into a magnetic bead-based aptasensor to detect C. parvum oocysts in recreational and drinking water samples. Using aptamer-conjugated magnetic beads, oocysts were first captured and concentrated prior to evaluation with the GNP-SPCE. With this method, increases in current intensity corresponding to a LOD of 50 oocysts was observed [68].

### 4.2. Fluorescence-Based Aptasensors

As with electrochemical aptasensors, fluorescence-based aptasensors exploit the highly dynamic nature of aptamers. In response to target binding, fluorophore-modified sequences undergo conformational and/or structural alterations. These changes in molecular proximity can affect the fluorescence or “Forster Resonance Energy Transfer (FRET)” between donor and acceptor fluorophores in close proximity, resulting in a quantifiable signal [103,104,105]. This concept has also been exploited with the growing use of quantum dot (QD) nanomaterials, whose fluorescence can be quenched and restored in the same fashion [106]. In light of this, quantum dots have been successfully integrated into aptasensors for *Pseudomonas aeruginosa* [107] and *Staphylococcus aureus* [108] detection. 

However, depending on the type of biosensor being used, target binding and concentration can be determined by detecting increases (“signal-on” mode) or decreases (“signal-off”) in signal strength. In the case of “signal-on” fluorescence-based aptasensors, an aptamer-conjugated fluorophore’s signal is initially suppressed by a neighbouring quenching molecule (Figure 4). Upon target binding, the aptamer’s conformational change liberates the fluorophore from the quencher’s FRET-based activity, restoring the fluorescence signal. Conversely, “signal-off” aptasensors are constructed in such a way that target binding results in a reduction in fluorescence emission.

#### 4.2.1. *Plasmodium* spp.

In combination with single layer molybdenum disulfide (MoS_2_) nanosheets, a fluorescein (FAM)-labeled aptamer specific for the pLDH malaria biomarker was used to yield a FRET-based aptasensor capable of pLDH detection in heterogeneous protein mixtures [59]. In this aptamer-based “capture-release” sensing assay, aptamer fluorescence was quenched upon adsorption to the MoS_2_ monolayer and recovered in response to pLDH binding. Based on the results of fluorescence recovery, the limit of detection for pLDH was estimated at approximately 550 pM, exceeding the requirements for clinical applications, where the mean level of pLDH in malaria-infected patients is estimated to be in the hundreds of nanomolar range [47]. However, the non-specific adsorption of both target and non-target biomolecules to the MoS_2_ surface was causing aptamer displacement, fluorescence recovery, and false positive results. To combat this, a blocking strategy using bovine serum albumin (BSA) to limit non-specific adsorption was successfully employed [60].

Metal nanoclusters (NCs) have gained recent attention in the field of diagnostics as their unique properties of intense fluorescence emission and photostability make them ideal biological probes [109]. Therefore, double-stranded DNA-scaffolded silver nanoclusters (AgNCs-dsDNA) were used in combination with a highly selective single-stranded DNA aptamer (2008s) to detect the well-established *Pf*LDH malaria biomarker at a concentration of 0.20 nM in buffer solution [48]. Similarly, sensitive *Pf*LDH detection was accomplished using aptamer-modified magnetic microparticles (MMPs) for capture and oligonucleotide-modified quantum dots (QDs) for detection. Furthermore, to increase detection sensitivity, the fluorescence signal was amplified using oligonucleotide-modified gold nanoparticles (AuNPs) to conjugate multiple QDs to each target antigen. Using this technology, the detection sensitivity of *Pf*LDH and *Pv*LDH was amplified from 0.5 fmole to 10 amole [49].

After showing that aptamer 2008s could still bind *Pf*LDH once integrated into a DNA nanostructure scaffold, a DNA nanobox, whose opening could be mediated by aptamer dependent *Pf*LDH recognition, was developed. By labelling the DNA nanobox with Cy3 and Cy5 fluorophores at strategic locations, a FRET-based assay was designed to monitor *Pf*LDH recognition and the subsequent change in box open/closed conformation. Upon target binding, aptamer conformational change caused the DNA nanobox to adopt an “open” conformation. In this scenario, the FRET signal was reduced due to the increased distance between Cy3 and Cy5 fluorophores. *Pf*LDH treatment resulted in an increase in box opening from 20% to 70% compared to the hLDH condition, where open conformation was stable at 20%. Furthermore, a reduction in FRET signal was determined in conditions with increasing concentrations of *Pf*LDH, corresponding to an estimated *K*_D_ value of 655 nM. While promising, this assay illustrated a significant decrease in aptamer sensitivity when complexed with the DNA nanobox, compared to that of the native aptamer, with a reported *K*_D_ of 42 nM [110].

As mentioned previously, *Pf*GDH has gathered interest as a potential biomarker of malaria infection due to some distinctive structural features differentiating it from its human counterpart HGDH. In light of this, a protein-based SELEX method was employed to develop a *Pf*GDH specific aptamer called NG3, whose binding affinity was subsequently approximated at 0.5 ± 0.04 µM using a circular dichroism (CD) assay. In order to detect *Pf*GDH, a fluorescent reporter assembly was constructed by chemically conjugating NG3 to carbon dots, a carbon-based nanomaterial with unique fluorescent properties. With this system, FRET-based detection of *Pf*GDH in human serum samples was accomplished via a protein-induced fluorescence enhancement (PIFE) phenomenon capable of detecting *Pf*GDH at concentrations as low as 2.85 nM. Additionally, when challenged with analogous malaria biomarkers *Pf*LDH and *Pf*HRP-II, signal to noise ratios were 8 and 4 times weaker, respectively (Figure 5) [111]. 

In a most recent work, the highly sensitive detection of *Pf*LDH in whole blood was made possible with an antibody-aptamer sandwich biosensor with gold nanoparticle substrates for fluorescent enhancement. In addition, a unique photochemical immobilization technique (PIT) was used for close-packing of antibodies on the AuNP surface. When treated with Cy5-labeled 2008s, antibody-captured *Pf*LDH was detected at femtomolar levels (LOD = 30 fM) in whole blood samples. Furthermore, *Pv*LDH could be captured by the anti-pLDH antibody layer of the sandwich biosensor but, due to the aptamer’s high specificity for *Pf*LDH, fluorescent readings mirrored those of negative controls [50]. 

#### 4.2.2. *Leishmania major*

Compared to malaria, the status of leishmaniasis as a neglected tropical disease is clearly illustrated by the reduced number of research articles geared towards aptamer-mediated diagnostics. Despite this, several aptamers have been developed against mitochondrial import receptors [112], the poly-A binding protein (PABP) [113], H2A and H3 histone proteins [114,115,116,117], and kinetoplastid membrane protein-11 (KMP-11) [61,90]. As such, a single fluorescence-based aptasensor was developed [62]. In their research, 10 cycles of whole-cell and magnetic bead-based protein SELEX were conducted to yield capture (LmWC-35R) and reporter (LmHSP-7b/11R) aptamers against *Leishmania major* promastigotes and recombinant hydrophilic surface protein (rHSP). In a sandwich type assay, capture aptamer-coated magnetic beads were used to stabilize rHSP while peroxidase conjugated reporter aptamers functioned to oxidize Amplex Ultra Red, yielding measurable amounts of fluorescent resorufin. This led to the development of a novel handheld fluorometric reader (FLASH) capable of detecting as little as 100 ng per 2 mL of MgCl_2_-extracted *L. major* promastigote protein from sandfly homogenate in under 1 h. While this method was successful in detecting promastigote protein extract in PBS samples, detection was not possible in 50% human serum [62]. Despite this, the FLASH field-portable assay may function as a useful surveillance tool to monitor the presence of *L. major* infected sandfly populations, helping to provide important epidemiological data. 

#### 4.2.3. *Cryptosporidium parvum*

The primary structure of any aptamer is vital in determining its crucial three-dimensional structures that allow target binding. However, in some cases, not all nucleotides contribute to their unique shape. This is the case with the U.S. Food and Drug Administration (USFDA) approved RNA aptamer against vascular endothelial growth factor (VEGF), whose binding sequence consists of only 27 nucleotides [118,119]. In light of this, the aforementioned R4-6 aptamer was truncated into two shortened sequences, designated Min_Crypto1 and Min_Crypto2, to improve the specificity and sensitivity of *C. parvum* detection in river and wastewater samples. More specifically, magnetic beads conjugated with fluorescein-labeled Min_Crypto2, which demonstrated the highest binding affinity, were used to capture and concentrate *C. parvum* oocysts prior to fluorescence intensity analysis. In spiked wastewater samples, a microplate fluorescence-based assay using the truncated aptamer could detect *C. parvum* oocysts at an LOD lower than the infective dose of 10 oocysts [66].

#### 4.2.4. *Toxoplasma gondii*

Current antibody-based methods of toxoplasmosis diagnosis are hindered by false-positive results due to the interference from various plasma proteins. Additionally, early phases of infection may yield false-negative results due to late seroconversion [120]. Therefore, specific capture and reporter aptamers were used alongside QDs to develop a more sensitive and specific quantum dots-labeled dual-aptasensor (Q-DAS) for anti-*Toxoplasma* IgG detection (Figure 6) [71]. After 10 cycles of SELEX with anti-toxoplasma IgG, two aptamers (TGA6 and TGA7) were identified and employed as probes in the biosensor. Firstly, the biotin-modified capture aptamer (TGA6) captures and immobilizes anti-toxoplasma IgG on a 96-well microplate. The QD-labeled detection aptamer (TGA7) is then introduced to yield a TGA6-IgG-TGA7 sandwich complex. Any emitted fluorescence from the QDs is subsequently measured to evaluate the concentration of captured anti-toxoplasma IgG. The described Q-DAS could detect the target antibody within the range of 0.5 to 500 IU with a limit of detection of 0.1 IU. In order to test the aptasensor’s specificity, it was challenged with an array of interfering agents and high-concentration blood-derived proteins. From this, the Q-DAS tool yielded high specificity with no false-positive reactions, an issue previously reported in traditional indirect fluorescent antibody tests (IFATs). With all of its strengths, however, the Q-DAS performance proved to be influenced by storage time, yielding changes in fluorescence variability over time. Finally, when compared to the Sabin-Fieldman immunoassay, considered the gold standard method for toxoplasma antibody detection, the Q-DAS could detect the target IgG with a specificity of 95.7% and sensitivity of 94.8% [71]. 

### 4.3. Colorimetric Aptasensors for Plasmodium Detection

Colorimetric biosensors achieve analyte detection through color changes that can be easily detected by the naked eye or optical detectors. In the last few decades, nanoparticles have been explored as colorimetric probes for the development of versatile biosensors due to their unique optical properties and abilities to induce color changes under different conditions. This is based on plasmonic effect, where the binding on an analyte to the particle can induce aggregation, resulting in interparticle surface plasmon coupling and color change [121,122]. More specifically, the use of gold nanoparticles (AuNPs) has catalyzed the advent of several colorimetric biosensors capable of highly sensitive analyte detection, including the first AuNP-based colorimetric aptasensor for potassium ion detection [20,123]. Since then, a variety of colorimetric aptasensors targeting infectious agents, including norovirus [124] influenza A virus [125], *E. coli* [126], and *Salmonella* [127], have been produced. In the field of parasitology, however, the application of this type of biosensor remains relatively unexplored and has so far been restricted to the *Plasmodium* genus.

Aptamer 2008s, mentioned previously in several aptasensors, was initially isolated and characterized following 20 rounds of magnetic bead-based SELEX against *Pf*LDH. With a *K*_D_ in the range of 20–50 nM, it was first conjugated to gold nanoparticles (AuNPs) to produce a colorimetric assay. In the presence of *Pf*LDH, AuNPs aggregation, induced by *Pf*LDH-aptamer binding, was visualized by transmission electron microscopy. In addition, the red color naturally emitted by naked AuNPs, monitored by absorbance at 520 nm, was lost under the aggregated state (Figure 7). Finally, the assay showed no specificity for human lactate dehydrogenase proteins hLDHA1 and hLDHB, and demonstrated an estimated LOD of 57 pg/µL [42]. 

The same aptamer was further exploited to develop a novel aptamer-tethered enzyme capture (APTEC) magnetic bead-based assay. Coupling the *Pf*LDH-catalyzed conversion of L-lactate to pyruvate with the reduction in nitrotetrazolium blue chloride (NTB) into a diformazan dye product, a measurable colorimetric response was produced upon aptamer-mediated capture of *Pf*LDH (Figure 8). The resulting aptasensor could detect *Pf*LDH with a LOD of 14 ± 6 fmol and correctly diagnose 12/15 light-microscopy confirmed *P. falciparum* infections [43]. Furthermore, 3D printing was used to integrate the principles of this colorimetric assay into paper-based syringe and magnetic bead-based well test point of care prototype devices [44]. While the proposed syringe test displayed a larger dynamic range and higher sensitivity than the well test, its requirement of additional processing steps for whole blood analysis makes it less friendly as a POC diagnostic tool. In addition, the cost of the well test, estimated at 0.36 USD, is significantly cheaper than the syringe test, estimated at 1.76 USD [44]. For further comparison, the average material cost of malaria RDTs has been estimated at around 1.51 USD [128].

Applications of the APTEC assay were further investigated to develop a mobile 3D-printed microfluidic biosensor capable of detecting *P. falciparum* from in vitro parasite cultures and malaria-infected patient samples. Following *Pf*LDH binding by a 2008s aptamer decorated micro-magnetic bead mobile phase, subsequent washing and development stages yielded a detectable colorimetric signal. When challenged with clinical samples from patients diagnosed with *P. falciparum* and *P. vivax* infection, the assay performed with a sensitivity of 90%, capable of detecting parasitemia as low as 0.01% [45]. 

*Plasmodium falciparum*-specific histidine-rich protein 2 (*Pf*HRP-2) is an important target antigen that has been used in many immunochromatographic malaria rapid diagnostic tests (mRTDs). However, in 2008, *P. falciparum* field isolates from the Peruvian Amazon tested negative with HRP2-based kits due to deletions of both *pfhrp2* and *pfhrp3* genes [129]. Since then, this issue has only worsened, with HRP2-deficient mutants having been isolated in Colombia, Brazil, and Bolivia, stressing the need for non-HRP2 RDTs [130,131]. Moreover, similar deletions have also been reported, though more scarcely, in Ethiopia [132] and India [133]. 

To address this issue, aptamer 2008s was challenged in ELONA and EMSA assays, displaying strong specificity and affinity to *Pf*LDH. In addition, 2008s showed no binding to *Pv*LDH, hLDH, or BSA. With these results, it was integrated into the APTEC assay to discriminate *Plasmodium falciparum* and *Plasmodium vivax* infections. Interestingly, when challenged with infected blood samples of both *Plasmodium* species, the 2008s-integrated APTEC assay could indeed differentiate the two cases. Unfortunately, false negative results in the *P. falciparum* infected cohort were found, potentially due to variance in the parasite’s metabolic stage [46].

Many colorimetric biosensors exploit the aggregative properties of AuNPs under conditions of high salt concentration. By fixing ssDNA aptamer P38 to the surface of AuNPs, aggregation was arrested and further controlled with the addition of *Pf*LDH. With increasing concentrations of *Pf*LDH, P38 dissociates from the AuNP surface, allowing for particle aggregation and a shift in color from red (dispersed AuNPs) to blue (aggregated AuNPs). Using this salt-mediated assay, a detection limit of 402 ± 40 pM was calculated. However, due to the disaggregation of AuNPs at higher protein concentrations during the salt-based assay, an alternative cationic surfactant-based assay was implemented. With the use of Benzalkonium chloride (BCK), the assay was capable of detecting *Pf*LDH at a more efficient limit of detection of 281 ± 11 pM [51]. Cationic polymers such as poly(diallyldimethylammonium chloride) (PDDA) and poly(allylamine hydrochloride) (PAH) have also exhibited significant advantages in relation to AuNP aggregation. In conjunction with ssDNA aptamer pL1, polymer-conjugated AuNPs were used to evaluate blood samples of malaria infected patients. In the absence of pLDH, polymer-conjugated AuNP aggregation is inhibited as pL1 binds and occupies PDDA or PAH. However, with increasing concentrations of pLDH, available pL1 diminishes as pL1-pLDH complexes are formed. Malaria positive cases therefore resulted in detectable color shifts from red (dispersed AuNPs) to blue (aggregated AuNPs). Respectively, as little as 74 (8.3 pM) and 92 (10.3 pM) *P. vivax* and *P. falciparum* parasites could be detected per µL of sample [57]. In a similar study, hexadecyltrimethylammonium bromide (CTAB) was employed as an aggregating agent to yield LODs of 1.25 pM and 2.94 pM for *Pv*LDH and *Pf*LDH, respectively. In human serum samples, however, the detection limits of *Pv*LDH and *Pv*LDH decreased to 10.17 pM and 13.54 pM, respectively [58]. 

### 4.4. Fiber Optic Aptasensor against Plasmodium falciparum

Compared to other sensing platforms, fiber optic biosensors (FOBS) are unique due to their optical-based method of signal transduction, which uses absorbance, reflectance, luminescence, refractive index, and light scattering to alter the signal for processing [134]. Many FOBS implement surface plasmon resonance (SPR) sensor configurations, a powerful technology that measures the refractive index of very thin layers of material adsorbed on a metal in response to biochemical interactions [135,136]. While this category of biosensor was initially developed using silica optical fibers, plastic optic fibers (POFs) offer a low-cost, robust, and high flexibility alternative [137,138]. Since their introduction, FOBS have demonstrated an impressive ability to detect a wide range of molecules using SPR and localized SPR transduction methods [134,139]. They have also successfully exploited a range of different MREs such as antibodies, molecular imprinted polymers (MIPs) and aptamers to produce sensitive and versatile detection platforms. Previously, fiber optic aptasensors have successfully detected environmental contaminants [140], antibiotics [141], disease biomarkers [142], and viruses [143].

While FOBS have been used previously to detect *Cryptosporidium parvum* [144], *Plasmodium falciparum* [145], and *Giardia lamblia* [146], the only aptamer-based FOB has been developed against *Plasmodium falciparum* to detect *Pf*GDH [53]. This aptasensor included a gold-sputtered U-bent POF functionalized with the previously mentioned *Pf*GDH-specific NG3 aptamer. A smartphone was integrated into the system to supply a light source and camera detector. As with previously described FOBS using SPR systems, this aptasensor functioned by probing the changes in refractive index in response to *Pf*GDH and NG3 interaction. The smartphone’s camera was used to take images from the fiber which were later processed by ImageJ software. Moreover, multiplex biosensing was made possible through multiple measuring channels.

In spiked buffer and diluted serum samples, the device detected *Pf*GDH concentrations as low as 264 nM and 352 nM, respectively. In tests of specificity, the smartphone-based POF aptasensor was challenged with analogous enzymes such as *Pf*LDH, HGDH, and *Pf*HRP-II. Although the system did respond to HGDH, the tool’s response to *Pf*GDH was 7–8 times higher than the interferents. Finally, the shelf-life of the aptasensor was assessed after 20 days in storage at 20–30 °C. After this time, the efficiency of the probe was 91.38% of the initial value [53].

### 4.5. Enzyme-Linked Oligonucleotide Assays (ELONA)

#### 4.5.1. *Trypanosoma cruzi*

During the acute phase of infection, *T. cruzi*, the causative agent of Chagas disease, can be detected in patient blood by microscopy and PCR. These techniques, however, become less effective during the chronic phase of infection, when parasitemia is at its lowest [147]. Conversely, indirect methods of detection, which target host anti-*Trypanosoma* antibodies, suffer from cross-reactivity issues in cases of co-infection and false negative diagnosis during the early stages of infection [148]. In order to overcome this, RNA aptamers specific for *T. cruzi* excreted secreted antigens (TESA), biomarkers of infection, were developed and implemented into an Enzyme-Linked Aptamer (ELA) assay to detect *T. cruzi* in infected mice (Figure 9). To monitor SELEX and the proportion of TESA-specific aptamers, mixtures of biotinylated sequences from various rounds of selection were challenged with TESA in the ELA assay. Sequencing and phylogenetic analysis of round 10 aptamers, which demonstrated the strongest ELA signals, indicated a convergence into 7 families representing 73.4% of the total number of clones sequenced. From this, an aptamer (Apt-L44) was challenged with the sera of mice infected with *T. cruzi* at various stages and could qualitatively detect infection as early as 7 dpi and as late as 230 dpi [63].

Aptamer sequences from round 10 of SELEX were subjected to an additional 11 rounds of selection to yield a 6-fold increase in the ELA assay TESA binding signal. In addition, aptamers from these pools produced higher signals at aptamer concentrations as low as 32.25 nM. Furthermore, sequences of the round 21 pool produced a signal 2.5 times higher than those of the round 10 pool when challenged with plasma of infected mice. Of the 7 sequences tested by ELA assay, Apt-29 yielded the strongest signals when challenged with *T. cruzi* TESA preparations. To monitor drug treatment efficacy during acute phase infection, the ELA assay was tested against the sera of Benznidazole-treated mice at 15 and 55 dpi. While a decrease in signal was shown from 15 to 55 dpi, all aptamers except Apt-1 displayed significantly higher levels of biomarker in the drug treated infected group compared to the drug treated non-infected control group, suggesting a reduction in parasitemia but failure to cure. In a similar fashion, the developed ELA assay was used to evaluate drug treatment in chronic cases of infection by using mouse sera at 130 and 170 dpi. In this case, 5 of 7 aptamers could detect significant differences in the level of biomarker between the infected drug treated group and the non-infected drug treated control [64].

#### 4.5.2. *Trichomonas vaginalis*

Current microscopy and culture-based methods of *Trichomonas vaginalis* detection suffer from issues of sensitivity. The *T. vaginalis* adhesion protein AP65 is a prominent adhesin located on the parasite’s cell surface. Responsible for mediating binding to host vaginal epithelial cells (VECs), it is also secreted into the extracellular environment [149]. In light of this, a microtiter plate SELEX (p-SELEX) method was used to enrich a library of DNA aptamers against AP65. Following selection, next generation sequencing (NGS) was performed and overrepresented sequences were screened by SPR. Aptamer AP65_A1 yielded the highest binding affinity with a *K*_D_ of 56 nM and was selected for an ELA assay. When challenged with AP65 protein, AP65_A1 demonstrated a *K*_D_ value of 1.057 nM and an LOD of 32 pM. However, when challenged with whole *T. vaginalis* cells in an ELA assay, AP65_A1 could detect no fewer than 8.3 × 10^3^ cells/mL. Finally, AP65_A1 was interrogated with various enteric and urogenital tract microorganisms to evaluate specificity and cross-reactivity. According to the ELA assay, the aptamer displayed high specificity for *T. vaginalis* cells with low cross-reactivity to *N. gonorrhoeae* and minimal cross-reactivity with the other 9 organisms tested [72].

#### 4.5.3. *Toxoplasma gondii*

The *Toxoplasma* ROP18 protein is secreted into the host cell upon invasion and has been classified as a key virulence factor in toxoplasmosis [150,151]. To detect *Toxoplasma gondii* in human serum samples, aptamers targeting ROP18 were produced using a protein SELEX strategy. Following 15 rounds of selection, Sanger sequencing identified two aptamers entitled AP001 and AP002, which represented 14.42% and 13.46% of the final enriched population, respectively. In order to construct an enzyme-linked aptamer assay (ELAA), both aptamers were biotin-labeled and used as biorecognition elements. In preliminary ELAA trials, the binding affinity of AP001 prevailed over AP002 with *K*_D_ values of 62.7 ± 17.27 nM and 97.7 ± 22.20 nM, respectively. Additionally, AP001 was capable of detecting rROP18 protein in serum samples at a concentration as low as 1.56 µg/mL. Finally, when challenged with patient samples, the direct ELAA platform positively identified 22.6% of toxoplasmosis cases and 60% of congenital infection cases [69]. 

SAG1 is a major surface antigen of *Toxoplasma* tachyzoites, serving an important role in host cell attachment [152,153]. As such, SAG1 has become an interesting target for laboratory diagnosis of toxoplasmosis. Following the work with ROP18, SELEX against recombinant SAG1 (r-SAG1) of *Toxoplasma* WH3 strain was performed using a synthetic oligonucleotide library containing Indole-dU, Phenol-dU, and Amine-dU modifications. From this, four aptamer candidates were identified by Next Generation Sequencing and screened using a direct enzyme-linked aptamer assay (DELAA). After challenging all four aptamers with native SAG1 (n-SAG1) from tachyzoite lysates, mouse sera of acute infection, and *Toxoplasma*-positive human sera, aptamer-2 proved to have the best performance with an estimated dissociation constant value of 41.57 ± 9.74 pM. The developed DELAA was further tested for its ability to monitor *Toxoplasma* infection in mice over time, detecting n-SAG1 as early as day 3 post infection. Finally, the DELAA was further evaluated with 15 positive and 35 negative human sera samples. In this test, the DELAA showcased a high sensitivity of 93.33% and specificity of 94.29% [70].

## 5. Discussion

Despite the many scientific advances made to improve global health, the ongoing emergence of infectious diseases highlights the increasingly complex relationship shared between humans, animals, and their natural environments. In an era defined by exponential advances in technology, increasing global travel, and rapid climatic changes, the fight against infectious disease is a reminder to collectively engage the broader issues of poverty, urbanization, and ecological collapse. This is true in the case of bacteria, viruses, and parasites, whose transmission to humans is achieved via insect vectors, ingestion of contaminated food or water, or by contact with infected animals and wildlife. With this in mind, the development of robust and effective tools for parasite detection in these various mediums is of high importance for monitoring, treating, and eradicating parasitic illnesses.

Although they are still in a nascent stage of development, the aptamer-based biosensors described herein are excellent examples of the wide range of functions that can be adopted by aptamers (Table 1). In addition to the aforementioned parasites in this text, aptamers targeting *Schistosoma japonicum* [73] and *Entamoeba histolytica* [154] have also been described. More specifically, fluorescently labeled aptamers specific for *S. japonicum* eggs, the causative agent of schistosomiasis pathology in humans, were produced to greatly facilitate tissue imaging and disease diagnosis [73]. Furthermore, therapeutic RNA aptamers capable of inhibiting *E. histolytica* proliferation, leading to cell death, have further illustrated the wide array of applications that aptamers possess in medical parasitology [154]. Finally, an aptamer selected against *Trypanosoma cruzi* could capture and concentrate live blood-borne trypomastigotes to facilitate PCR-based detection in blood [155].

Today, despite the obvious advantages of aptamer-based biosensors, parasite detection continues to rely heavily on antibody-based testing. While antibodies have ultimately changed the landscape of diagnostics for the better, their limitations, inherent to their complex protein structures, have recently brought aptamers closer to the forefront of diagnostics research. In order to reach commercial availability, however, improving the performance of aptamer-based detection platforms remains a priority. Luckily, this can be accomplished with the incorporation of chemical modifications to improve aptamer affinity, specificity, and stability. This is especially valuable in the context of parasitic illnesses endemic to tropical countries, whose warm and humid climates demand robust and heat-stable tests. Along with chemical additions, aptamer performance can also be improved using steps of negative selection throughout the SELEX process. By doing so, aptamers capable of binding various targets are eliminated, ensuring the selection of sequences that uniquely bind the target of interest. Aptamer performance can also be improved by using high quality targets. In the field of parasitology, this means using purified proteins that carry the appropriate post-translational modifications to ensure high affinity binding or, in the case of whole-cell SELEX, using live parasites so that native proteins are targeted. When used effectively, these methods can reduce the possibility of selecting aptamers with low specificity capable of cross-reactivity.

Additionally, high-throughput sequencing (HTS) technologies and bioinformatics software should be more frequently considered, as they offer possibilities that have seldom been exploited in aptamer research. While these tools have been used endlessly in other domains, their ability to analyze sequence diversity following each round of SELEX offers researchers a glance into the complex dynamics of aptamer selection. Together, these valuable tools allow us to evaluate the rate at which different sequences and sequence families evolve, granting us the knowledge to more easily identify potential aptamer candidates and explore different mutation avenues.

The current review outlines aptamers as promising tools of parasite detection in humans as well as the various vehicles and vectors that transmit them. Unfortunately, aptamer-based research against parasites remains mostly neglected, with the majority of work surrounding *Plasmodium* spp. Despite this, their additional success in detecting other pathogenic organisms highlights their potential to have a real-world impact on the lives of many.

## Figures and Tables

**Figure 1 sensors-23-00562-f001:**
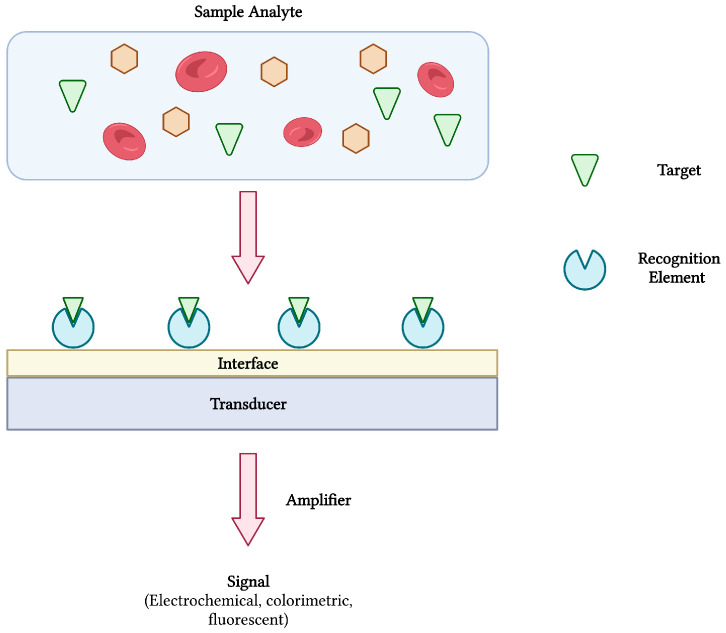
The schematic diagram of a biosensor and its components. The interaction between target analyte and recognition element is converted and amplified into a measurable signal. Adapted with permission from [17]. Created with BioRender.com (accessed on 19 December 2021).

**Figure 2 sensors-23-00562-f002:**
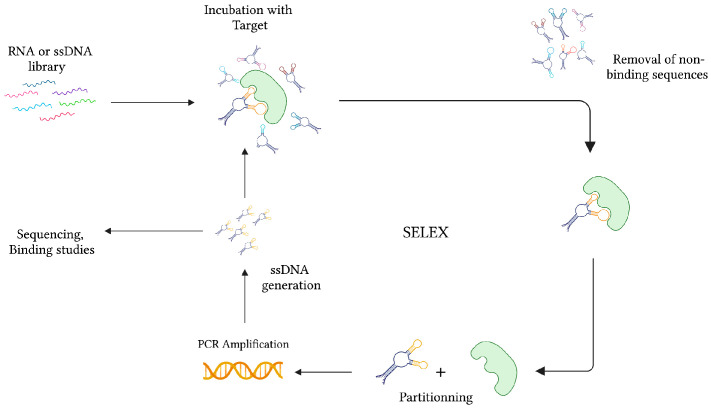
The graphical representation of the SELEX strategy for ssDNA aptamer selection. In the case of RNA aptamer selection, the ssDNA library is first amplified into dsDNA before transcription for RNA synthesis. After collecting the target-specific sequences, RNA is amplified in two consecutive steps of reverse-transcription and transcription. Created with BioRender.com (accessed on 19 December 2021).

**Figure 3 sensors-23-00562-f003:**
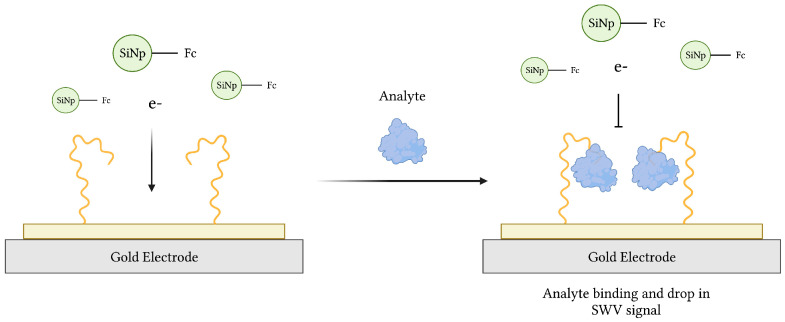
The schematic illustration of an impedimetric electrochemical-based aptasensor. Aptamers immobilized on the surface of a gold electrode act as MREs to capture analyte from a sample. The resulting decrease in electron transfer from ferrocene (Fc)-modified silicon nanoparticles (Fc-SiNPs) to the electrode surface is proportional to the concentration of analyte. Adapted with permission from Ref. [79]. 2015, American Chemical Society. Created with BioRender.com (accessed on 19 December 2021).

**Figure 4 sensors-23-00562-f004:**
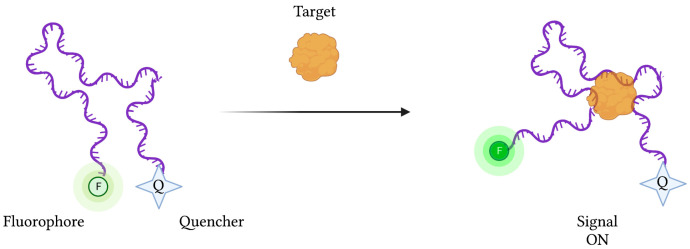
The schematic illustration of a generic “signal-on” Fluorescence-based aptasensor. Biorecognition of the target analyte by the aptamer induces a change in conformation, resulting in a displacement of the fluorophore away from the quencher, resulting in fluorescence emission. Created with BioRender.com (accessed on 19 December 2021).

**Figure 5 sensors-23-00562-f005:**
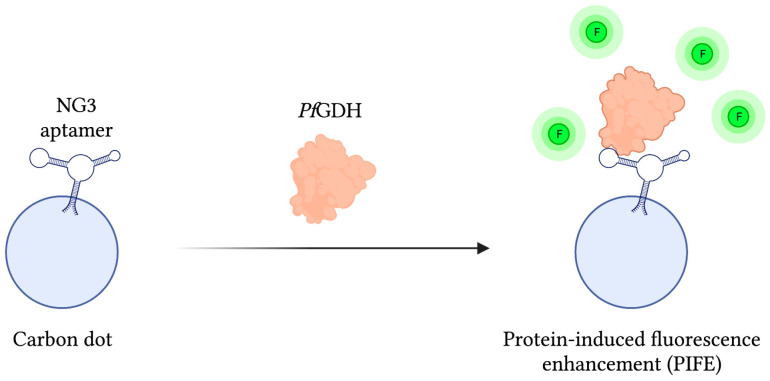
The schematic illustration of the Fluorescence-based aptasensor for *Plasmodium falciparum* GDH detection described by Singh et al. [111]. The fluorescence intensity (F) of the carbon dot-aptamer assembly is enhanced by their interaction with *Pf*GDH. Adapted with permission from Ref. [111]. 2018, American Chemical Society. Created with BioRender.com (accessed on 19 December 2021).

**Figure 6 sensors-23-00562-f006:**
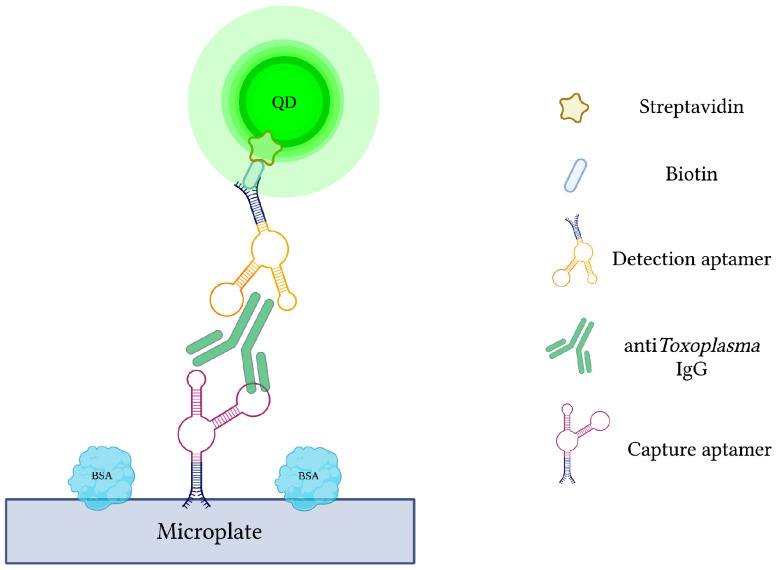
The schematic illustration of the Quantum dots-labeled dual-aptasensor (Q-DAS) for anti-*Toxoplasma* IgG detection. Capture and reporter aptamers bind to different epitopes of anti-*Toxoplasma* IgG. Capture aptamers immobilize the IgG while aptamer-labeled quantum dots emit a measurable fluorescence. Adapted with permission from Ref. [71]. 2013, American Chemical Society. Created with BioRender.com (accessed on 19 December 2021).

**Figure 7 sensors-23-00562-f007:**
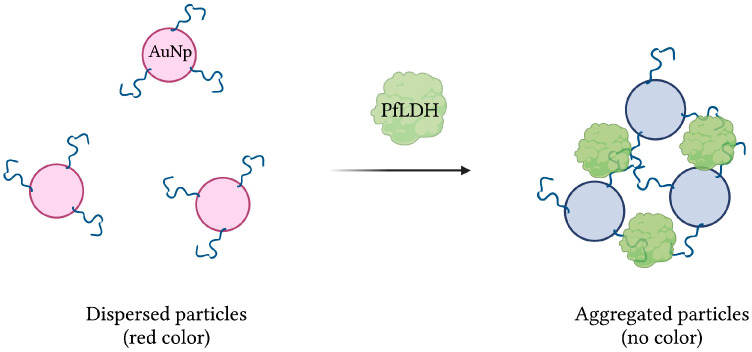
The schematic illustration of the colorimetric-based aptasensor for *Plasmodium falciparum* LDH detection. Adapted with permission from [42]. Aptamer-conjugated AuNps aggregate in the presence of *Pf*LDH, resulting in a detectable loss of their red color. Created with BioRender.com (accessed on 19 December 2021).

**Figure 8 sensors-23-00562-f008:**
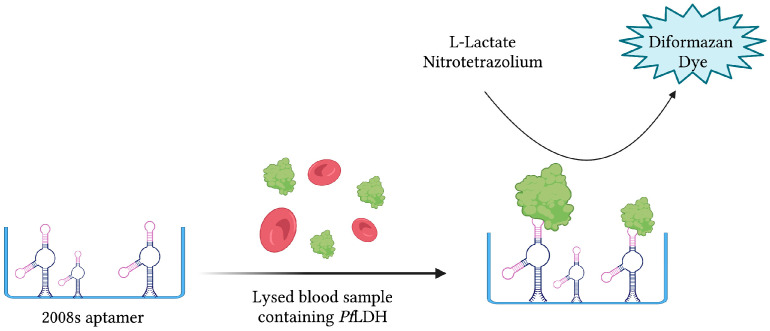
The schematic illustration of the colorimetric-based aptasensor for *Plasmodium falciparum* LDH detection described by Dirkzwager et al. [44]. *Pf*LDH is captured and immobilized by 2008s aptamers. A *Pf*LDH-mediated catalytic reaction yields a detectable color change proportional to its concentration. Adapted with permission from Ref. [44]. 2016, American Chemical Society. Created with BioRender.com (accessed on 19 December 2021).

**Figure 9 sensors-23-00562-f009:**
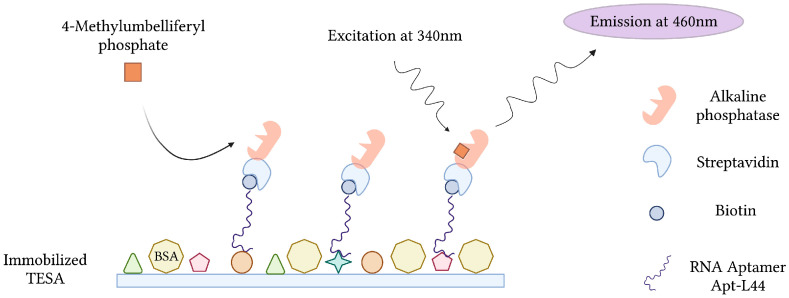
The schematic illustration of the Enzyme-Linked Aptamer (ELA) assay for the detection of *T. cruzi* excreted secreted antigens (TESA) described by Nagarkatti et al. [63]. Created with BioRender.com (accessed on 19 December 2021).

**Table 1 sensors-23-00562-t001:** The summary of the aptamers and aptasensors developed against various parasites.

Target Organism		Target Protein	Aptamer Name	Aptasensor Type	Limit of Detection (LOD)	Reference
***Plasmodium* spp.**	*P. falciparum*	*Pf*LDH	2008s	Electrochemical	0.84 pM	[40]
					1.0 pM	[41]
				Optical/Colorimetric	57 pg/uL	[42]
				APTEC Optical/Colorimetric	14 ± 6 fmol	[43]
					5 ng/mL (syringe test) 50 ng/mL (well test)	[44]
					0.001% parasitemia	[45]
					N/A	[46]
				FRET-based	N/A	[47]
				Fluorescence	0.20 nM	[48]
					10 amol	[49]
					30 fM	[50]
			P38	Electrochemical	0.5 fM	[51]
				Optical/Colorimetric	281 ± 11 pM	[52]
		PfGDH	NG3	Optical Fiber	352 pM	[53]
				Electrochemical	0.77 pM	[52]
				FRET-based	2.85 nM	[54]
		HRP-II	B4	Electrochemical	3.15 pM	[55]
	*P.vivax* *P. falciparum*	pLDH (PfLDH/PvLDH)	pL1	Electrochemical	108.5 fM (*Pv*LDH) 120.1 fM (*Pf*LDH)	[56]
				Optical/Colorimetric	8.3 pM (*Pv*LDH) 10.3 pM (*Pf*LDH)	[57]
					1.25 pM (*Pv*LDH) 2.94 (*Pf*LDH)	[58]
		pLDH	N/A	FRET-based	550 pM	[59,60]
***Leishmania* spp.**	*L. infantum*	LiKMP-11	SELK10	Electrochemical	2.27 uM	[61]
	*L. major*	rHSP	LmWC-35R LmHSP-7b/11R	Fluorescence	50 ng/mL in sandfly homogenate	[62]
***Trypanosoma* spp.**	*T. cruzi*	TESA	Apt-L44	ELA assay	N/A	[63]
			Apt-29/Apt-71			[64]
	*T. brucei*	VSG	cl57	Electrochemical	10.0 pM	[65]
***Cryptosporidium* spp.**	*C. parvum*	Whole oocysts	Min_Crypto1 Min_Crypto2	Fluorescence	10 whole oocysts in wastewater	[66]
			R4-6	Electrochemical	100 whole oocysts in fruit juice homogenate	[67]
					50 oocysts in recreational and drinking water samples	[68]
***Toxoplasma* spp.**	*T. gondii*	ROP18 protein	AP001 and AP002	Colorimetric	1.56 ug/mL in human serum	[69]
		SAG1 Protein	aptamer-2		N/A	[70]
		Antitoxoplasma IgG	TGA6 and TGA7	Fluorescence	0.1 IU	[71]
***Trichomonas* spp.**	*T. vaginalis*	AP65	AP65_A1	ELA assay	32 pM	[72]
***Schistosoma* spp.**	*S. japonicum*	*S. japonicum* whole eggs	LC-6 LC-15	N/A	N/A	[73]

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
