# Peer review of "Aptamer-Based Technologies for Parasite Detection"

_sensors, 2023, doi:10.3390/s23020562_

Round 1
Reviewer 1 Report
This review offers an overview of the aptamer-based technologies for parasite detection. The review started from the introduction of biosensor, and then focused on the aptasensor based on different transducer methods for detection of different parasites by discussing the performance and mechanisms. The review concludes with potential directions for future research in the development of aptamer-based technologies for parasite detection. I think that the topic is interesting and should attract a broad range of readers. However, some revisions are needed before it can be accepted for publication.
Major comments:
1) Abstract: should be rewrote. For example, the authors claimed that “this review details the methods used to generate parasite-specific aptamers as well as the performance and mechanisms of various aptamer-based biosensors.” I am not sure the methods used to generate parasite-specific aptamers are introduced. SELEX is a common way to screen aptamer. Moreover, the authors just gave a very short introduction to SELEX.
2) 1.1 Biosensors as diagnostic tools: some descriptions are very jumping. For example, the first appearance of antibody and aptamer. I think in this part the concept of biosensors should be given. Such as the major classes of biorecognition materials, antibody, aptamer and so on. Please refer to Biosensors and Bioelectronics, 2019, 144, 111693 to complete the concept of biosensor.
3) I strongly suggested to delete the subsections in introduction part because three of them are not the in parallel relationship. By the way, there are typo mistakes because there are three subsection 1.1.
4) For a review paper, most of references are old than five years ago. It will be helpful that more references can be added into the introduction part to show the current researches in the biosensor field, such as ACS Sens, 2021, 6, 3367; Environ. Sci. Technol. 2022, 56, 14350; Environment International, 2021, 146, 106181.
5) Section 2. Aptasensors for parasite detection. For my understanding, this part should provide the sensing principle and representative examples of aptasensor for parasite detection. And the content should be closely related with Table 1, summarization of aptamers and aptasensors developed against various parasites. subsection 2.1 focused on electrochemical aptasensor and tried to give a general introduction to the principle of electrochemical aptasensor before moving into different parasites. However, I think the general introduction from lines 136-151 is not accurate. For electrochemical aptasensor, it can be current type, voltage type, and resistance type. So I suggested to delete lines 136-151 and directly introduced different aptasensors for different parasites. In each subsection, one typical example can be shown with the figure to explain the sensing principle.
6) Figure 2. It is totally wrong. The sensing area of a screening-printing electrode is the gold or carbon area with white background. However, the authors put all aptamers on the isolation area.
7) Subsection 2.2 and 2.3: same as the 2.1, the general introduction part is not so broad to cover the statements. The details under each subsections should be added with more beautiful and easy-to-understand figures to show the typical examples about parasite detections.
8) If there is only 2.4.1, the authors should delete 2.4.1 and all contents belong to 2.4
9) Subsection 2.4 and 2.5: copy more beautiful and easy-to-understand figures to show the typical examples about parasite detections again.
10) The references in Section 2 should be closely related with those presented in Table 1.
Specific comments:
11) Line 67, we usually call “environmental health” as “environmental monitoring”.
12) All parasite names should be italicized.
13) Page 1, no figure caption should be added for TOC.
Author Response
Hello,
Firstly, I want to thank you for your expert opinions and rigid review of our paper. I have done my best to make corrections in response to your recommendations. See attachment for details

Reviewer 2 Report
This review presents several aptamer-based technologies for parasite detection. These biosensors include electrochemical aptasensors, fluorescence-based aptasensors, colorimetric aptasensors, optical fiber aptasensors and Enzyme-linked oligonucleotide assays. In general, the paper is clear and compact, and is suitable for publication in Sensors journal after being revised.
1 More pictures or tables should be added to make the paper more readable.
2 The subheading of sections 2.3 and 2.4 are not enough. Are there any other parasites that could be detected by colorimetric aptasensors or optical fiber aptasensors?
3 Please check the subtitle No. in Introduction section.
Author Response
Hello,
Many thanks for you comments and feedback on our article. We have made some changes in response.
1) a total of 3 figures have been added
2) Unfortunately for section 2.4 this is the only case of a fiber optic aptasensor used in parasite detection that I have seen. Section 2.5 has been condensed into what is now "4.3"
3) Indeed we have reorganized the introduction section into different sections
Many thanks again for you helpful feedback

Round 2
Reviewer 1 Report
I think the authors have reponded to my concerns one by one. Some obvious flaws have been corrected. Some minor revisions should be corrected.
1. The authors should update the references. Because newly added references are not present in the reference list.
2. Most figures are obscure. Please make sure they can meet the publication requirement.
Author Response
Dear Reviewer,
Many thanks for the additional comments. Please see attached our responses to your comments.
